# Optimization of the Subcritical Butane Extraction of Tung Oil and Its Mechanism Analysis

**Zicheng Zhao** [1,2]**, Xudong Liu** [2]**, Desheng Kang** [3]**, Zhihong Xiao** [2]**, Wenzheng Dong** [1]**, Qiquan Lin** [1,*]
**and Aihua Zhang** [2,*]

1   College of Mechanical Engineering, Xiangtan University, Xiangtan 411105, China;
    z1053514332@163.com (Z.Z.); wzdong@xtu.edu.cn (W.D.)
2   State Key Laboratory of Utilization of Woody Oil Resource, Hunan Academy of Forestry,
    Changsha 410004, China; lxd1995621@163.com (X.L.); xzhh1015@163.com (Z.X.)
3   Hunan Xiangchun Agricultural Technology Co., Ltd., Changsha 410004, China; jjz4459151@163.com
*   Correspondence: xtulqq@163.com (Q.L.); zhangaihua909@163.com (A.Z.)

**Abstract:** Tung oil is an important dry grease. In order to overcome the deficiencies of traditional processes in oil production, the preparation of tung oil was carried out by the butane-subcritical method. A response surface optimization experiment was carried out based on Design-Expert software, and the best process parameters were obtained. The extraction temperature was 42.98 °C, the extraction time was 43.77 min, the particle size of the raw material was 38.88 mesh, and the oil yield of tung oil under this condition reached 67.437%. The fatty acid composition of tung oil was analyzed by Gas Chromatography-Mass Spectrometry (GC-MS): the content of α-oleostearic acid was 74.99%, linoleic acid content was 8.83%, oleic acid content was 7.42%, palmitic acid content was 2.02%, and stearic acid content was 4.35%. Through the analysis of the oil sample obtained, five indicators showed that the process of obtaining oil products met the requirements of the national standard. By simulating the subcritical n-butane/tung oil dissolution equilibrium model, the miscible dynamic equilibrium of tung oil in subcritical n-butane was studied at temperatures in the range of 35–50 °C and an equilibrium time of 40 min, and the kinetic equations of oil extraction at different temperatures were obtained, with a coefficient of determination ($R^2$) greater than 0.99. The oil extraction rate was up to $67.12 \pm 0.05\%$ under optimal extraction conditions through the optimization of univariate and response surface experimental design. Using 1stOpt data processing software, the data of tung oil extraction rate at different times were fitted, and it was found that the Patricelli model accurately elucidated the kinetic process of tung oil extraction through subcritical n-butane, with $R^2$ greater than 0.99.

**Keywords:** subcritical; response surface method; tung oil; extraction

## 1. Introduction

Tung oil is a vegetable oil that is composed of Tung fordii (Hemsl.) when Airy Shaw seeds are pressed and refined. Tung oil has been widely used in the synthesis of paints, coatings, and ink, and in construction, machinery, chemicals, vehicles and ships, electronics, and other industries [1–3]. If tung oil can be fully utilized for deep processing and comprehensive utilization, it will have positive economic and social benefits.

In recent years, with the rapid development of China's economy, the demand for tung oil in all walks of life has increased [4,5]. Tung oleic acid contains three conjugated double bonds; the system was named cis, trans, trans-9, 11, 13-octadecatrienoic acid, and has lively chemical properties under the action of heavy metals, water, and sulfuric acid, or light properties under the action of morphological transformation and triggered polymerization reaction [6–8]. Therefore, different preparation processes have direct and different impacts on the quality of tung oil [9].

At present, there are two main techniques for extracting tung oil: pressing and dipping. The pressing process required the roasting of tung seed raw materials, which may easily cause local overheating and the β transformation of tung oil. In the stage of machine oil production, due to the high throughput of mechanical extrusion, friction will also produce a high temperature area ($\geq$120 °C), directly acting on the tung oil, which causes the fixation and β transformation of tung oil and affects the product quality [10,11]. Compared to other technologies, the leaching method has the advantages of high extraction efficiency, large production scale, low production cost, and a high degree of automation. The leaching method mainly uses 6# solvent oil to extract tung oil from tung seeds using the principle of similar compatibility, but 6# solvent is mainly obtained by fossil crude oil extraction, and the sulfur content is high, which can cause deterioration of the tung oil. Although the extraction efficiency is high and it is easy to apply on a large scale and the residual oil in the oil is basically less than 1%, the temperature of the oil and cake desolvation process can be up to 200 °C, which can destroy the heat sensitive components in the oil and produce benzopyrene, trans fatty acid Polycyclic aromatic hydrocarbons, glycidyl esters, and other harmful components [12]. Subcritical refers to the material state of some compounds in the form of fluids under conditions where the temperature is higher than their boiling point but below the critical temperature, and the pressure is lower than its critical pressure, which has the advantages of enhanced molecular diffusion performance, fast mass transfer speed, and strong permeability and solubility of non-polar, medium weak polarity and strong polar substances in natural products [13]. Compared with other oil-making processes, it has the advantages of low residual oil rate (<1%) and low extraction temperature (30~50 °C). Its oil-making principle is based on the mutual solubility caused by the similar polarity between solvent and oil molecules, and the low-temperature oil making process can protect the heat sensitive components in the material, which is convenient for subsequent quality improvement and efficiency development and the utilization of cake. The process needs very little additional heat energy supplemented in the process of extracting oil, which significantly reduces energy consumption. Moreover, the price of n-c4 is lower than that of 6# solvent, and the loss of solvent in the extraction process is very small, so it can be recycled many times [14].

In this study, the feasibility analysis of tung oil prepared by tung seeds is discussed in relation to subcritical butane technology, and the process optimization of the factors affected the extraction rate of tung oil is carried out using a Box–Behnken experimental design, so as to obtain the best process parameters. At the same time, the physical and chemical properties, fatty acid composition, and functional group changes of tung oil products are investigated and provide a reference for the preparation of high-quality tung oil and the future development and utilization of tung oil resources.

## 2. Materials and Methods

### 2.1. Experimental Materials

#### 2.1.1. Instruments

CBE-100 L Subcritical Fluid Extraction Equipment (Henan Subcritical Biotechnology Co., Ltd., Anyang, China), Scion SQ Gas phase mass spectrometry (Brook Dalton, Leipzig Germany), an RA-200 CNC standard inspection screening machine (Shanghai Ruang Electromechanical Technology Co., Ltd., Shanghai, China), a VGT-2013 QTD Desktop ultrasonic cleaner (Guangdong Good Ultrasonic Co., Ltd., Ghuangzhou, China), a XFB-200 Laboratory Universal Crusher (Jishou Zhongcheng Pharmaceutical Machinery Factory, Jishou, China), and aMS304TS-02 Electronic Balance (METTLER TOLEDO International Trading Shanghai, Shanghai, China) were utilized in this work.

#### 2.1.2. Reagents

Tung Seed (Chenzhou Guosheng Bioenergy Co., Ltd., Chenzhou, China), butane (99.99%, Changsha Xinxiang Gas Chemical Co., Ltd., Changsha, China), methanol (AR grade, Tianjin Kemio Chemical Reagent Co., Ltd., Tianjin, China), and sodium hydrox-

ide (AR grade, Sinopharm Chemical Reagent Co., Ltd., Shanghai, China) were used in this work.

*2.2. Research Methodology*

2.2.1. Subcritical Extraction of Tung Oil

The dried tung seeds were dehulled and purified, crushed by a universal crusher, and then screened and graded by a CNC standard inspection and screening machine to obtain the required particle size. The quantitative material was accurately weighed into an extraction kettle, under the temperature and pressure set for investigation. Butane was used as a solvent to extract the fat-soluble tung oil by subcritical process; the solvent in the tung meal and tung oil was evaporated under reduced pressure, low-temperature tung meal and extracted tung oil were obtained, the oil was weighed, and the oil rate was calculated. The solvent gas was compressed and condensed by the compressor and then liquefied for subsequent recycling [15].

2.2.2. Oil Content Index Detection

The oil content index detection referred to was GB/T 14488.1-2008. An amount of 10 g of pre-dried, crushed, and other treated materials was accurately weighed through the pre-treatment packaging into the HX-S-04 Sowell extraction device. Petroleum ether or ether was selected as the extraction solvent, and water bath temperature was set at 85 °C, with a reflux time of 8 h. Finally, the extraction solution was dried with the solvent, the weight of the grease was calculated by the subtraction method, and the oil content was calculated by the comparison of the quality of the raw material, as in Formula (1).

$$w_0 = \frac{m_1}{m_0} \times 100\% \tag{1}$$

where:

$w_0$—Oil content of the sample/%;
$m_0$—Sample weight/g;
$m_1$—Extract grease weight/g.

2.2.3. Study of the Tung Oil-Subcritical n-Butane Two-Phase Solubility Equilibrium Model

The tung oil, large-hole filter paper, and a glass dish were dried at 70 °C to a constant weight. The quantitative tung oil was accurately weighed and transferred to the glass dish and the beaker opening was closed with the large-hole filter paper, and this was then moved to the extraction kettle of the CBE-XL type SPE experimental equipment. The equipment agitation was set on slow, the control metering device was passed into the set amount of n-butane (8L), and the temperature and pressure was modulated. The process lasted for a set time, as in Figure 1, for the specific operation. After completion, the beaker was removed and the remaining tung oil was placed into the oven at 70 °C to be dried to a constant weight; the difference in tung oil was calculated as the solubility under this condition, as in Formula (2).

$$\Delta m = m_0 - m_1 \tag{2}$$

where:

$\Delta m$—Loss of weight of tung oil/g;
$m_0$—Initial mass weight/g;
$m_1$—Remaining weight/g.

$$S = \frac{\Delta m \times \rho_T}{\rho_{20}} \tag{3}$$

where:

S—Solubility of tung oil/$(g \cdot L^{-1})$;

$\rho_T$—Density of liquid n-butane at reaction temperature/(kg·L$^{-1}$);
$\rho_{20}$—The density of liquid n-butane at 20 °C was 0.579/(kg·L$^{-1}$).

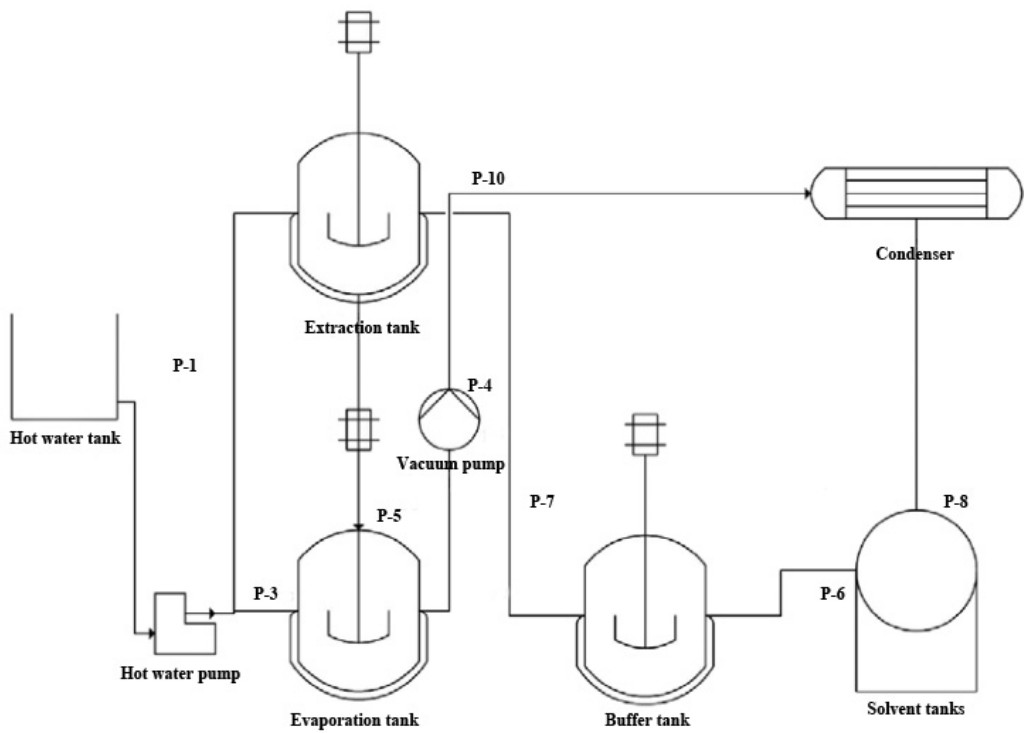

**Figure 1.** Process flow chart of subcritical extraction.

2.2.4. Kinetic Study of Subcritical n-butane Fluid Extraction of Tung Oil

(1)    Subcritical extraction to produce oils

The pretreated tung fruit granules were placed into the extraction bag and weighed. The extraction bag was then transferred to the extraction kettle, and the subcritical n-butane fluid extraction was carried out with the equipment closed, as in Section 2.2.1. The calculation method for extraction rate is shown in Formula (4).

$$w(\%) = \frac{m_0 - m_1}{m_0} \times 100\% \tag{4}$$

where:

w—The extraction rate of tung oil/%;
$m_0$—The weight of the material before extraction/g;
$m_1$—The weight of the material after extraction/g.

(2)    Dynamic model selection

Three kinetic models with different theoretical bases were selected for subcritical n-butane fluid extraction: mass transfer model of Fick's law, secondary reaction kinetic model, and the empirical model. The process of extraction and mass transfer is described by reaction kinetics, and the selected model is shown in Table 1.

**Table 1.** Dynamic model of oil production by subcritical butane fluid extraction.

| Dynamic Model | Model Expression | Model Parameter |
|---|---|---|
| Fick's Law [16] | $\frac{R_t}{R_e} = 1 - Ae^{-B_1 t}$ | A, $B_1$, $R_e$ |
| Secondary reaction kinetics [17] | $\frac{dc}{dt} = k_1(c - c_\infty)^2$ | $c_\infty$, $k_1 c_\infty$ |
| Empirical model | $c = \frac{t}{k_1 + k_2 t}$ | $k_1$, $k_2$ |

### 2.2.5. Experimental Design for Subcritical Extraction of Oils

(1)  Univariate experiments investigated the influence of different factor variables on the extraction effect: material particle size/mesh (10,20,30,40,50), the temperature of extraction/°C (25,30,35,40,45), the time of extraction/min (20,30,40,50,60), ratio of feed–liquid (1:1,1:1.5,1:2,1:2.5,1:3), and the number of extractions (1,2,3,4,5).

(2)  Response surface optimization experiment: Based on the first part of Section 2.2.5, the influence of significant factors and their interactions on the extraction effect were further investigated, and the experimental design adopted the BBD principle to design L17 (3,3) for RSM optimization.

### 2.2.6. Temperament Analysis

Chromatographic conditions [18]: FID detector, OV-1 column (30.0 m × 0.25 mm × 0.25 μm), helium as a carrier gas with a flow rate of 10. 0 mL/min. The injection volume was 1.0 μL, inlet temperature was 260.0 °C, and ion chamber temperature was 240.0 °C.

Procedure heating conditions: initial temperature 50.0 °C (hold for 2.0 min), at a heating rate of 10.0 °C/min to 190.0 °C (hold for 10.0 min), at a heating rate of 5.0 °C/min to 240.0 °C (hold for 20.0 min).

Mass spectrometry conditions: Scion SQ single quadrupole mass spectrometer, electron bombardment (EI) ion source, electron energy of 70.0 eV, quadrupole temperature of 150.0 °C, ion source temperature of 230.0°C. Mass scan range of 33 to 350 amu.

### 2.2.7. Detection of Physical and Chemical Properties of Tung Oil

Refer to the national standard GB 8277-1987 for the determination of the physical and chemical indicators of tung oil [19].

## 3. Results and Analysis

### 3.1. Study on the Tung Oil-Subcritical n-Butane Two-Phase Solubility Equilibrium Model

The simulation system in Figure 2a can be divided into three regions of A, B, and C; the concentration distribution of tung oil is greatest in the A region, which is greater than the C region and the B region. The mass transferred movement of lipid molecules starts from the initial contact between the tung oil and the subcritical n-butane phases; the A region was transmitted to the C region and then from the C region to the B region, and n-butane was transmitted from the B region to the C region and then from the C region to the A region. After a certain period of mass transfer movement, the three regions of A, B, and C reached a dynamic equilibrium.

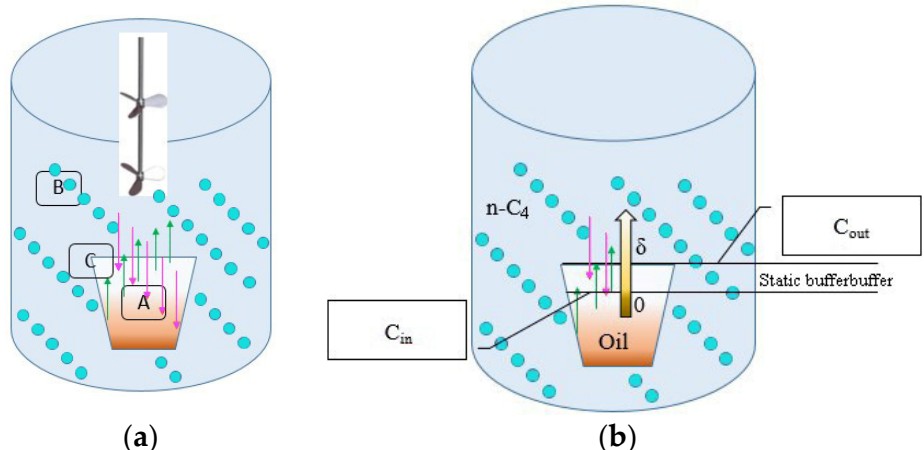

(**a**)　　　　　　　　　　　(**b**)

**Figure 2.** Model diagram. (**a**) Schematic diagram of simulated two-phase miscible equilibrium: A—Tung oil area, B—n-butane region, C—sealing. (**b**) Schematic diagram of one-dimensional steady-state diffusion model.

### 3.1.1. Study of Dynamic Miscible Equilibrium Duration

The equilibrium mass transfer time directly affected the accuracy of the miscibility measurement results, and in order to ensure the achievement of the system phase equilibrium, a reasonable mass transfer time is particularly important. An excessively long mass transfer time will not only affect the accuracy of the results but will also cause energy loss and increase unnecessary burdens. Mass transfer times that are too short will directly affect the accuracy of miscibility, rendering the experiments meaningless. Therefore, under the conditions of 30 °C, 40 °C, and 50 °C, the influence of mass transfer time on miscibility was investigated, and the results are shown in Figure 3.

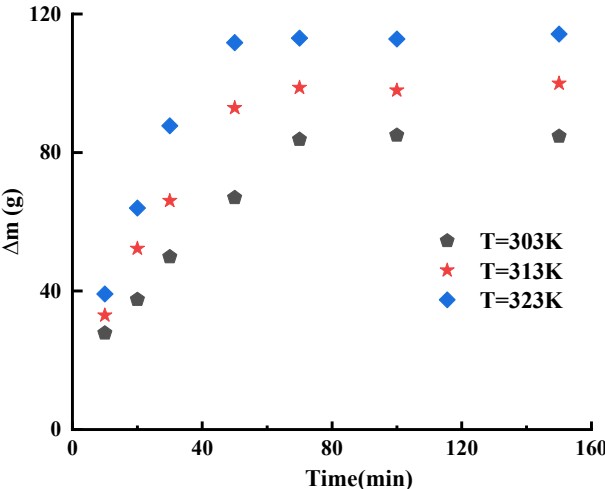

**Figure 3.** Changes in the weight of tung oil in butane fluid.

As shown in Figure 3, the mass of dissolved tung oil in the n-butane fluid at different temperatures showed a trend of first increasing and then gradually flattening over time, and the relative standard deviation (RSD) is used to show the stability of the experimental process data; the calculation method of RSD is shown in Formula (5).

$$RSD = \frac{S}{\overline{x}} \times 100\% = \frac{\sqrt{\sum_{i=1}^{n} \frac{(x_i - \overline{x})^2}{n-1}}}{\overline{x}} \times 100\% \tag{5}$$

where:

n—The number of data collected;
$x_i$—The data value for the data point;
$\overline{x}$—Collected average of the data.

Each of the three adjacent pieces of data were taken as a group, and the RSD for each set of data was calculated according to Formula (5). When the RSD is below 5.0%, it can be considered that the data generated by the experimental process are stable; that is, the miscible mass transfer process reaches dynamic equilibrium.

As can be seen from Figures 3 and 4, under the temperature condition of 30 °C, the miscible diffusion of tung oil reached equilibrium after 60 min. Under the conditions of 40 °C and 50 °C, the time of dissolution equilibrium of tung oil was around 50 min, which shows that an increase in temperature can make tung oil and solvent reach miscible equilibrium faster. This is because the temperature increase can speed up the mass transfer rate and increase the level of molecular motility, thereby shortening the time for dissolution to reach equilibrium. In order to accurately measure the miscibility of tung oil within the temperature range (30~50 °C), the appropriate extraction equilibrium time was selected as 60 min.

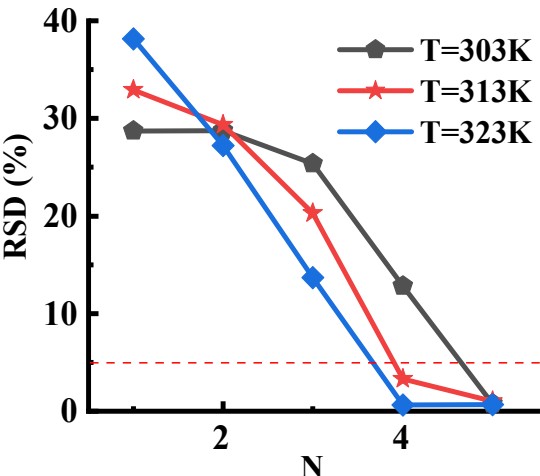

**Figure 4.** Collected group data RSD.

3.1.2. Statistical Validation of Solubility

The accuracy of this protocol was tested by repeating the experiments, and the accuracy and stability of the experimental data were measured by relative standard deviation (Formula (5)). Investigation process conditions: reaction temperature of 30 °C, 1 L of solvent, three groups of tung oil/n-butane miscible data (85.31 g/L, 86.22 g/L, 86.83 g/L). Standard deviation RSD was less than 1%, so the results of this experiment are accurate and reliable.

3.1.3. Miscible Model Validation

As can be seen from Figure 2a, the n-butane molecules and the tung oil molecules transfer mass to each other through the relatively static buffer C and then enter each other's systems in a gradient diffusion, and finally the respective concentrations reach the miscible equilibrium state under set conditions.

Based on the nature of mass transfer diffusion, a one-dimensional steady-state diffusion model of tung oil in n-butane fluid was established, and the schematic diagram is shown in Figure 2b.

$$\frac{\partial c}{\partial t} = \frac{\partial}{\partial y}\left(D\frac{\partial c}{\partial y}\right) = 0 \tag{6}$$

then:

$$\frac{\partial c}{\partial y} = a(Fixed\ Value) \tag{7}$$

Pair of y points:

$$c = ay + b(Fixed\ Value) \tag{8}$$

Substitute boundary conditions:

$$\begin{cases} C|y = 0 = C_{in} \\ C|y = \delta = C_{out} \end{cases} \tag{9}$$

where $C_{in}$ and $C_{out}$ are the mass concentrations (g/L) of tung oil, which are substituted to obtain:

$$a = \frac{C_{out} - C_{in}}{\delta} \tag{10}$$

$$b = C_{in} \tag{11}$$

$$c = \frac{C_{out} - C_{in}}{\delta}y + C_{in} \tag{12}$$

Fick's first law of diffusion is adopted:

$$J = -D\frac{dc}{dy} \tag{13}$$

The mass flow rate through the static buffer area of A per unit time is:

$$\frac{dm}{dt} = JA = -DA\frac{dc}{dy} = -\frac{DA}{\delta}\left(\frac{m}{v_{out}} - \frac{m_{in}}{v_{in}}\right) \tag{14}$$

When the concentration of tung oil in the A region is consistent in $C_A$ and the $C_B$ in area B is consistent, and it is set as a constant term, the constant term in Formula (14) is replaced by P and R.

$$\frac{DAm_{in}}{\delta v_{in}} = P \tag{15}$$

$$\frac{DA}{\delta v_{out}} = R \tag{16}$$

Equation (14) can be reduced to:

$$\frac{dm}{dt} = P - Rm \tag{17}$$

On t-points:

$$m = C_1 e^{-C_2 t} + C_3 \tag{18}$$

After the above deduction, the relationship between the dissolved mass, m, and the elapsed time, t, within a certain period of time, is shown in Equation (18). Using the nonlinear fitting function of origin 2019 to fit the experimental data of Figure 3, $y = A_1 e^{-t/t_1} + y_0$, and the result is shown in Figure 5.

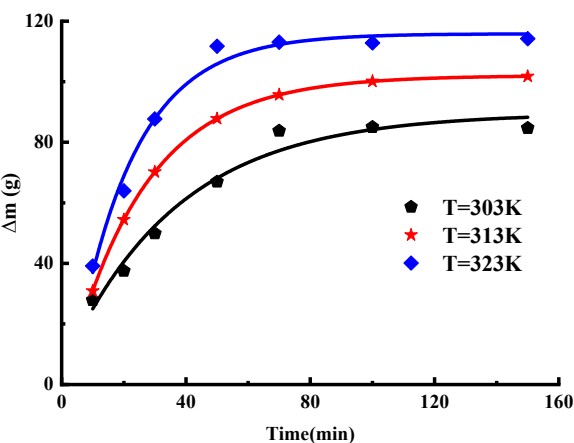

**Figure 5.** Experimental data fitting results.

The effect of the fit was judged by the mean squared difference and the coefficient of determination ($R^2$), and the fitting result is shown in Table 2.

**Table 2.** Model fitting results.

| T (°C) | $y_0$ | $A_1$ | $t_1$ | $R_2$ | F |
|---|---|---|---|---|---|
| 30 | 89.61408 | −85.10845 | 36.32023 | 0.97643 | $1.72 \times 10^{-5}$ |
| 40 | 102.02705 | −106.28086 | 24.88152 | 0.98397 | $5.82 \times 10^{-6}$ |
| 50 | 115.83204 | −132.20575 | 19.34857 | 0.98672 | $2.38 \times 10^{-6}$ |

As can be seen from Table 2, the coefficient of determination of the time result, $R^2$, was large ($R^2$ was greater than 0.95), and the F values are significant (F was less than 0.05%). The test of variance showed that the calculated values of the model do not differ significantly from the experimental data.

3.1.4. Miscible and Interrelated

The miscibility value of temperature 30~50 °C was determined, and the results are shown in Table 3.

**Table 3.** Experimental data of solubility.

| T (K) | $\Delta m^*$ | $\rho_T$ (kg·L$^{-1}$) | S (g·L$^{-1}$) |
|-------|--------------|------------------------|----------------|
| 303 | 84.7 | 0.568 | 83.09 |
| 313 | 99.9 | 0.556 | 95.93 |
| 323 | 114.21 | 0.543 | 107.1 |

The data correlation method of miscible equilibrium adopted the association model method [20]. The brief derivation process is as follows:
1 mol A and k mol B form a 1 mol conjugate ABk, as:

$$A + kB \rightarrow AB_k \tag{19}$$

There are equilibrium constants:

$$K = \frac{[AB_k]}{[A][B]^k} \tag{20}$$

The logarithm is used to obtain:

$$lnK + ln[A] + kln[B] = ln[AB_k] \tag{21}$$

where [A], [B], and [AB$_k$] represented the equilibrium molar concentrations of solutes, solvents, and conjugates, respectively.

$$[AB_k] = \frac{c}{M_A + kM_B} \tag{22}$$

Derived from Formulas (19)–(22):

$$c = d^k exp\left(\frac{a}{T} + b\right) \tag{23}$$

where d is the fluid density and a, b, and k are empirical constants, respectively.

Formula (26) correlated the miscibility with the n-butane fluid density, operating temperature, and other factors, and linearly reverts the temperature and density values in Table 3, as can be seen in Formula (24). The regression coefficient of 0.99893 indicated a high linear correlation in the range of 30 to 50 °C.

$$\rho_T = -0.00125T + 0.94692 \tag{24}$$

Formula (24) are carried into Formula (26):

$$c = (-0.00125T + 0.94692)^k exp\left(\frac{a}{T} + b\right) \tag{25}$$

The experimental data were returned with Formula (25) to obtain the regression equation:

$$S = (-0.00125T + 0.94692)^{38.83} exp\left(\frac{-9816.15}{T} + 58.76\right) \tag{26}$$

Because a $= \frac{H}{R} < 0$, this demonstrates that the dissolution of tung oil in subcritical n-butane was an exothermic process.

The comparison of experimental values with the calculated values of Formula (26) is shown in Figure 6.

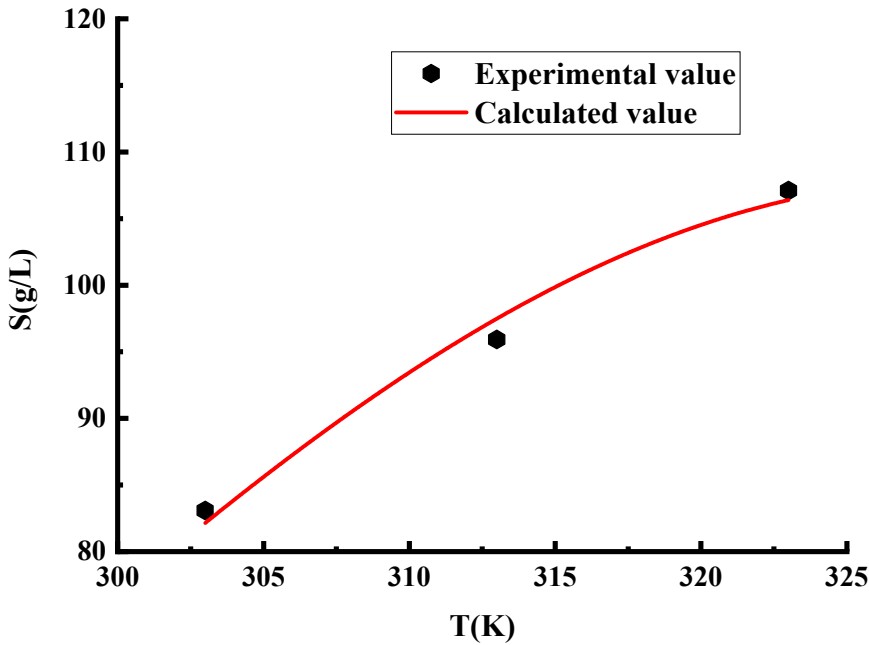

**Figure 6.** Comparison between calculated value and fitting value.

As can be seen from Figure 6, both the experimental and the calculated values fall on a straight line. The average relative error (ARE%) of the experimental values was 1.07%, and the maximum relative error (RE%) was 1.12%, so it is shown that in the range of 30 to 50 °C, the miscibility in the calculation of tung oil and subcritical n-butane by Formula (9) has good accuracy, which is of great significance for the temperature guidance selected for this experiment.

*3.2. Study on the Oil Production Process of Subcritical n-Butane Fluid Extraction*

3.2.1. One-Way Experiments

(1)　Particle size factors affect the extraction rate

Experimental conditions: extraction temperature of 45 °C, feed-to-liquid ratio of 1:1.5, extraction time of 40 min.

The experimental results are shown in Figure 7a. It can be seen that the particle size of the material was within 40 mesh and the extraction rate of tung oil was proportional to the mesh number; the particle size of the material was greater than 40 mesh, and the extraction rate of tung oil is inversely proportional to the mesh number. In summary, the extraction rate is the highest when the particle size is 40 mesh. This is mainly due to the fact that the tung grain was too large to easily cause the extraction barrier to increase. The cell wall was not sufficiently broken, which is not conducive to the diffusion of tung oil, but also reduced the contact area between the solvent and the material, resulting in a decrease in the oil yield. At the same time, a too small tung seed grain will lead to an increase in the specific surface area of the material, increasing the adsorption capacity, which is not conducive to the extraction of tung oil, and too small a particle size will make the material and solvent difficult to separate, contaminating the pipeline system.

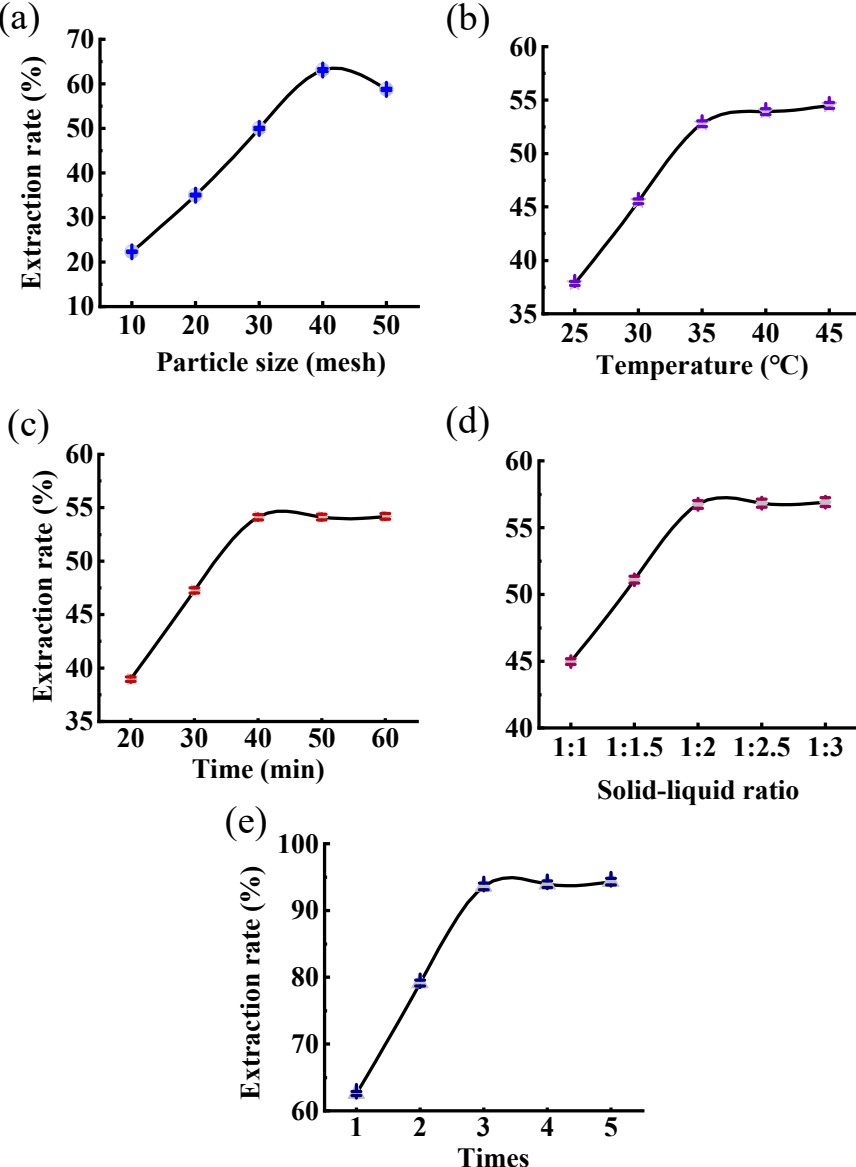

**Figure 7.** Effect of factors on extraction rate. (**a**) Particle size, (**b**) Temperature, (**c**) Time, (**d**) Solid-liquid ratio, (**e**) Times.

(2)    The effect of extraction temperature on extraction rate

Experimental conditions: material-liquid ratio of 1:1.5, material particle size of 40 mesh, extraction time of 40 min.

The experimental results are shown in Figure 7b. It can be seen that in the process of subcritical n-butane fluid extraction and oil production, temperature increase (25~35 °C) can effectively improve the extraction rate of oil, which is mainly because the viscosity of tung oil is relatively high. The temperature directly affected the diffusion coefficient of the solvent and the tung oil itself, and the temperature increase will promote the diffusion ability of tung oil and the dissolution performance of the solvent, but the higher temperature will caused solvent gasification, resulted in the gasification of n-butane molecules in the extraction kettle, so that the actual volume of n-butane solvent in the subcritical state is reduced, resulting in a decrease in solvent density.

A reduced contact area between the material and the extraction solvent is also not conducive to the extraction of tung oil, and the previous experiment found that the temperature being too high also directly affected the quality of tung oil, resulted in a darker color, which is not conducive to the extraction of tung oil. Therefore, the optimization

range for the extraction temperature factor in the subsequent response surface experiment is 30~40 °C.

(3)    Effect of extraction time on extraction rate

Experimental conditions: extraction temperature was 45 °C, feed–liquid ratio was 1:1.5, material particle size was 40 mesh.

The experimental results are shown in Figure 7c. The extraction reached a dynamic equilibrium when the extraction time was 40 min. This is mainly due to the subcritical state of the extraction fluid and tung seed particles after contact, according to the gradient difference in the concentration of oil and fat inside and outside the tung seed to produce a driving force. The oil molecules in the material cells form a high concentration due to n-butane solvent rapid diffusion, so the extraction rate has been significantly improved through the similar phasic principle of the solvent extracting tung oil into the solvent. With the extension of time, the concentration difference decreased, the n-butane solvent in the grease molecules and the material cells in the grease molecule diffusion movement achieved dynamic equilibrium, and extraction efficiency was reduced, until dynamic equilibrium was achieved. If the extraction time is extended, the extraction rate cannot be increased, so the investigation range of 30 to 50 min for subsequent experiments is appropriate.

(4)    Effect of feed-to-liquid ratio on tung oil extraction rate

Experimental conditions: extraction temperature of 45 °C, material particle size of 40 mesh, extraction time of 40 min.

The experimental results are shown in Figure 7d. It can be seen that in the subcritical extraction process, in the material–liquid ratio from 1:1 to 1:25, the extraction rate has been significantly improved, which is due to the small amount of solvent affecting the mass transfer of oil molecules in the material. As the material-to-liquid ratio increases, it will increased the concentration difference of the diffusion of oil molecules in the extraction process, which is conducive to the improvement of the extraction rate. When the feed-to-liquid ratio is further increased, the increase in the diffusion balance of the oil molecules is not as great as the increase in the feed-to-liquid ratio, and it will also cause the cost of subsequent dissolution to increase. Therefore, it is appropriate to select the investigation range of the feed–liquid ratio of 1:1.5~1:2.5 for subsequent experiments.

(5)    Effect of extraction time on the extraction rate of tung oil

Experimental conditions: extraction temperature of 45 °C, material particle size of 40 mesh, extraction time of 40 min, feed–liquid ratio of 1:1.5.

The experimental results are shown in Figure 7e. It can be seen that the extraction rate of tung oil increased with the increase in the number of extractions, and after three extractions, the increase in the extraction rate of oil and fat was significantly reduced, which indicates that the extraction rate of tung oil after three extractions under the current process conditions reaches a peak of 94.16%. The result of increasing the number of extractions is that the extraction time and feed-to-liquid ratio will increase, which will not only cause a waste of solvents and prolong the time of desolation, but also increase the cost of process production. Therefore, it is advisable to select the number of extractions as three times, and for the independence of the factors, the factor of the number of extractions is no longer considered in the subsequent response surface experimental design.

### 3.2.2. RSM Protocol Results

According to the experimental investigation results of Section 3.2.1, the independent variables in A—the time of extraction, B—the temperature of extraction, and C—the particle size of material, were selected as the independent variables in the RSM experiment, and the Y-extraction rate of tung oil was the response value of the experimental results. RSM optimization was performed using the BBD module in Design-Expert 12.0, and the experimental design was L17(3,3).

The Box–Behnken data processing and analysis of the experimental results are shown in Table 4. The quadratic regression equation model between the three factors of extraction time (A/min), extraction temperature (B/°C), and material particle size (C) during the preparation of tung oil is shown in Formula (27).

$$Y = 67.60 + 0.3275 \times A + 0.6850 \times B - 0.5775 \times C - 0.085 \times AB + 0.355 \times AC - 1.62 \times BC - 1.93 \times A^2 - 3.00 \times B^2 - 4.47 \times C^2 \quad (27)$$

**Table 4.** Box–Behnken experimental design and results.

| Experiment Number | Factor | | | Y (%) |
|---|---|---|---|---|
| | A/Min | B/°C | C/Mesh | |
| 1 | 30 | 43 | 50 | 59.92 |
| 2 | 30 | 48 | 40 | 62.62 |
| 3 | 40 | 38 | 30 | 57.89 |
| 4 | 50 | 48 | 40 | 62.82 |
| 5 | 40 | 38 | 50 | 59.73 |
| 6 | 40 | 43 | 40 | 67.52 |
| 7 | 50 | 43 | 30 | 61.76 |
| 8 | 40 | 48 | 30 | 63.78 |
| 9 | 40 | 43 | 40 | 67.97 |
| 10 | 50 | 43 | 50 | 61.57 |
| 11 | 40 | 43 | 40 | 67.13 |
| 12 | 50 | 38 | 40 | 62.89 |
| 13 | 30 | 38 | 40 | 62.35 |
| 14 | 40 | 43 | 40 | 67.72 |
| 15 | 30 | 43 | 30 | 61.53 |
| 16 | 40 | 48 | 50 | 59.12 |
| 17 | 40 | 43 | 40 | 67.64 |

It can be seen from Table 5 that the model P was less than 0.0001, $R^2$ (deciding coefficient) was 0.9776, and $R^2$adj (adjustment coefficient) was 0.9487, indicated that the fitting model was significant, and the fitting degree was high. In addition, the response value of tung oil extracted from subcritical n-butane fluid can be accurately predicted under different conditions.

**Table 5.** Response surface regression model analysis of variance.

| Source Model | Sum of Squares | df | Mean Square | F-Value | *p*-Value |
|---|---|---|---|---|---|
| Model | 169.89 | 9 | 18.88 | 33.90 | <0.0001 |
| *A* | 0.8581 | 1 | 0.8581 | 1.54 | 0.2545 |
| *B* | 3.75 | 1 | 3.75 | 6.74 | 0.0356 |
| *C* | 2.67 | 1 | 2.67 | 4.79 | 0.0648 |
| *AB* | 0.0289 | 1 | 0.0289 | 0.0519 | 0.8263 |
| *AC* | 0.5041 | 1 | 0.5041 | 0.9052 | 0.3731 |
| *BC* | 10.56 | 1 | 10.56 | 18.97 | 0.0033 |
| $A^2$ | 15.69 | 1 | 15.69 | 28.18 | 0.0011 |
| $B^2$ | 37.78 | 1 | 37.78 | 67.84 | <0.0001 |
| $C^2$ | 84.15 | 1 | 84.15 | 151.1 | <0.0001 |
| Residual | 3.90 | 7 | 0.5569 | | |
| *Lack of fit* | 3.52 | 3 | 1.17 | 12.34 | 0.0172 |
| *Pure Error* | 0.3801 | 4 | 0.0950 | | |
| Cor Total | 173.79 | 16 | | | |
| | $R^2 = 0.9776$ | | $R^2_{adj} = 0.9487$ | | |

According to the regression equation, the parameters of subcritical extraction of tung oil and the effect of the two-by-two interaction between the three factors of extraction time

(A/min), extraction temperature (B/°C), and material particle size (C) on the extraction rate of the experimental results were examined, as shown in Figure 8.

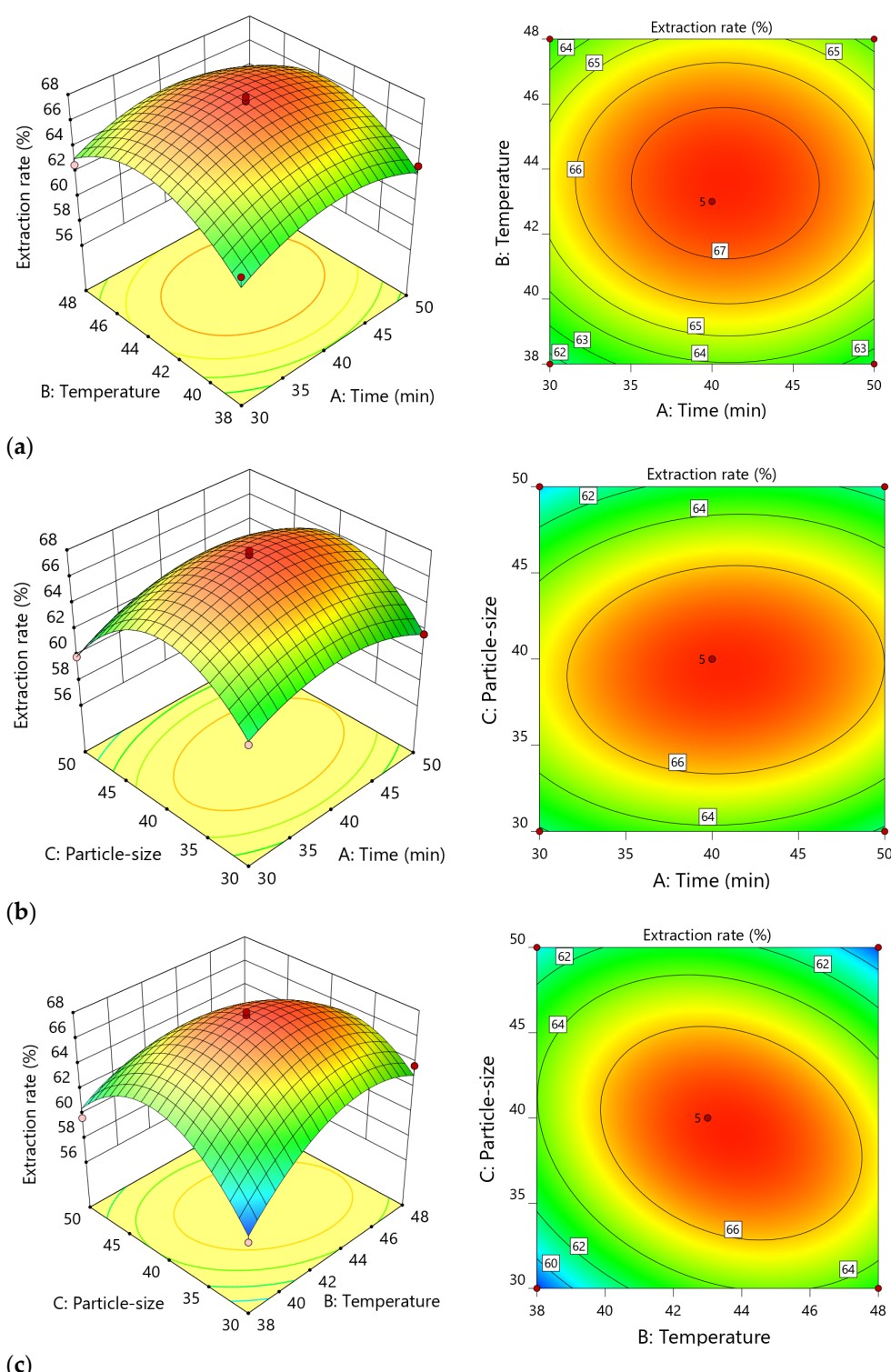

**Figure 8.** Response surface and contour maps of two factors interacting to influence the conversion rate. (**a**) Interaction between time/temperature; (**b**) Interaction between time/particle-size; (**c**) Interaction between temperature/particle-size.

As can be seen from Figure 8, the response surface interaction 3D graph and contour plot showed the highest point of the interaction, and the values of the highest point are within the scope of the investigation, indicating that the model is true and reliable.

Further analysis showed that the extraction time and particle size of the two factors were very significant, followed by the extraction temperature. After analyzing the variance of the regression model of the response surface of Table 4, the best parameters were determined to be an extraction time of 43.77 min, an extraction temperature of 42.98 °C, a particle size of the material of 38.88 mesh, and a predicted extraction rate of 67.437%. According to the actual situation, it is adjusted to the extraction time (44 min), extraction temperature (43 °C), material particle size (39), and the three parallel experimental values after the optimization of the response surface were 67.12 ± 0.05% (shown in Table 6), which was 0.31 ± 0.05% different from the predicted values, indicating that the model was true and reliable, and has a certain guiding significance for practical operation.

**Table 6.** Comparison between predicted optimal values and actual values of SFE.

| Experimental Condition | A/Min | B/°C | C/Mesh | Y/% |
|---|---|---|---|---|
| Optimal in the model | 43.77 | 42.98 | 38.88 | 67.437% |
| Optimal in practice | 44 | 43 | 39 | 67.12 ± 0.05% |

3.2.3. Kinetic Study of Subcritical n-Butane Fluid Extraction of Tung Oil

Meziane et al. [21] studied the extraction mechanism of sunflower seeds using the Patricelli model and found that there were two ways of extraction: free lipid extraction and combined oil diffusion extraction, of which free oil extraction was dominant, and the model is relatively mature. Our study directly cites this model in Formula (28) for subsequent kinetic studies.

$$Y_t = Y_e^w \left(1 - e^{-B_w t}\right) + Y_e^d \left(1 - e^{-B_d t}\right) \tag{28}$$

where

$Y_t$—The extraction rate of tung oil at a certain point in time/(g/100 g);

$Y_e$—Extraction equilibrium rate;

$Y_e^w$—Extraction equilibrium of free lipid extraction/(g/100 g);

$Y_e^d$—Combined lipid diffusion extraction equilibrium rate/(g/100 g);

$B_w$—Free grease mass transfer coefficient/(min$^{-1}$);

$B_d$—The mass transfer coefficient of the combined oil/(min$^{-1}$).

It can be seen from Figure 9 that, under the same temperature conditions, the extraction rate of subcritical n-butane fluid extraction showed an increasing trend with the time increased and within 20 min of the initial stage. The extraction rate of tung oil increased rapidly, which is due to the high content of oil molecules in the material and the fact that the gradient difference of diffusion is relatively large, so it is beneficial to the extraction rate. In the time that followed, the extraction rate gradually flattened out. At the same time, with the increase in temperature, the extraction rate also showed an upward trend, which is due to the fact that a rise in temperature can increase the rate of movement of fluid and grease molecules, which is beneficial for the extraction of grease molecules.

Using the 1stOpt data processing software to match the data of tung oil extraction rate at different temperatures and different times and using origin software for plotting, it can be seen from Figure 10 that the Patricelli model accurately expounds the kinetic process of subcritical n-butane extraction. The fitting curve is shown in Figure 10 in the red solid line, and the model parameters are shown in Table 7.

$$Y_t = 67.41 \left(1 - e^{-0.44t}\right) + 8.24 \left(1 - e^{-0.00231t}\right)$$

$$Y_t = 66.39 \left(1 - e^{-0.53t}\right) + 7.66 \left(1 - e^{-0.00212t}\right)$$

$$Y_t = 65.97 \left(1 - e^{-0.53t}\right) + 8.05 \left(1 - e^{-0.00212t}\right)$$

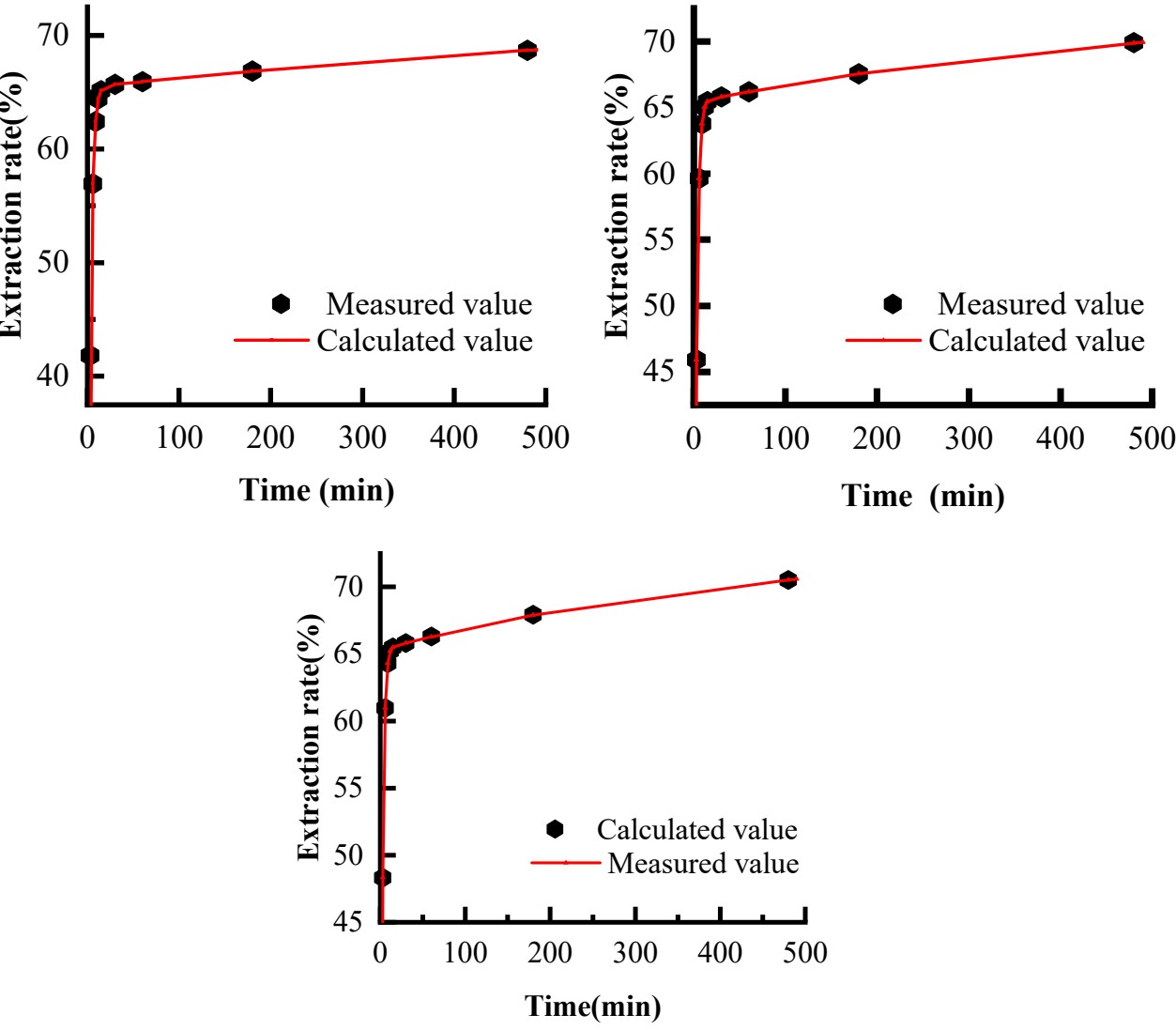

**Figure 9.** Kinetic curve at different extraction temperatures.

**Table 7.** Fitting parameters of the Patricelli model in the extraction process of Tung oil.

| T/°C | Mass Transfer Coefficient/Min$^{-1}$ | | Equilibrium Yield/% | | | Coefficient of Determination |
|------|-----------|--------------|------------|-----------|----------|----------------|
| **T** | **B$_W$** | **B$_d$** | **Y$_e^W$** | **Y$_e^d$** | **Y$_e$** | **R$^2$** |
| 35 | 0.44 | 0.00231 | 67.41 | 8.24 | 73.15 | 0.9923 |
| 40 | 0.53 | 0.00212 | 66.39 | 7.66 | 73.8 | 0.9914 |
| 45 | 0.548 | 0.00301 | 65.97 | 8.05 | 74.26 | 0.996 |

The grease extraction process described by the Patricelli model is divided into two parts: one part is that after the material is pretreated, the free grease that adhered to or was adsorbed inside the material or between the material particles was generated, and the extraction rate of this part of the grease was very high, and the trend of a steep rise can be seen in the figure. The second is the binding grease inside the material, which was in the cell wall; this part requires the diffusion movement of the grease molecules to achieve the extraction of the grease, which has a lower extraction rate due to the increased obstruction.

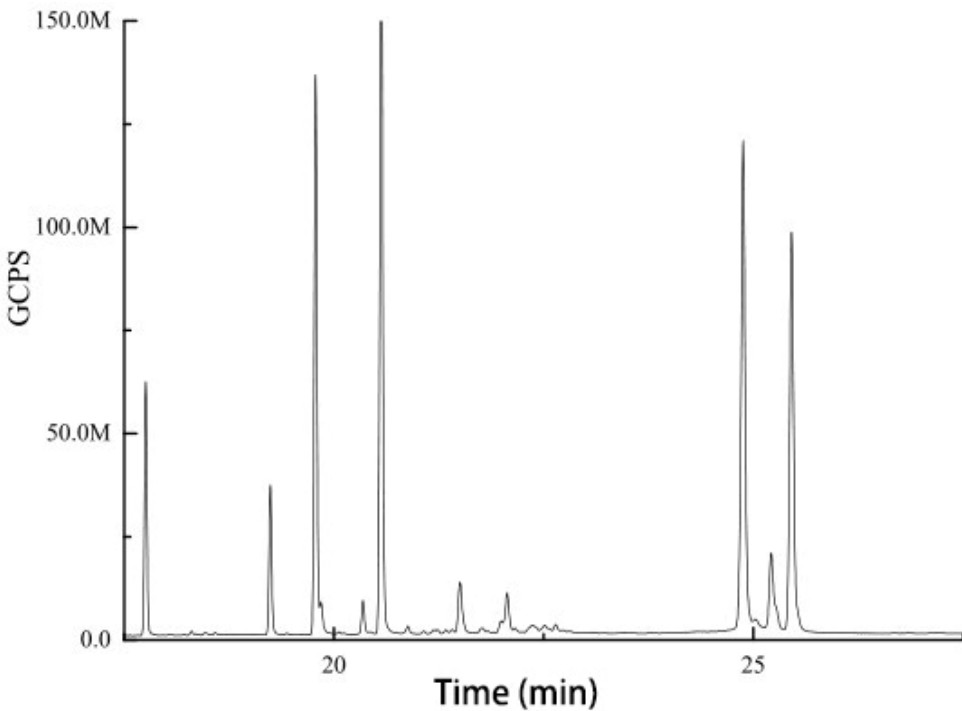

**Figure 10.** GC-MS of tung oil prepared by butane-subcritical process extraction.

### 3.3. The GC-MS Spectrum of Tung Oil

Fatty acid fraction analysis of tung oil prepared by butane-subcritical process extraction was performed by Gas Chromatography–Mass Spectrometry (GC-MS), and the results are shown in Figure 10. It can be seen from Figure 10 that the components of tung oil fatty acids prepared by the butane-subcritical process contain α-berlic acid of 74.99%, linoleic acid of 8.83%, oleic acid of 7.42%, palmitic acid of 2.02%, and stearic acid of 4.35%, and there is almost no difference from the distribution of fatty acids in the references [22,23], which verified the feasibility of this process.

### 3.4. Physicochemical Index of Tung Oil

In order to investigate the quality of tung oil prepared from the butane-subcritical process, five of its indicators were tested with reference to the national standard GB 8277-1987 of tung oil, and the results are shown in Table 8. From Table 8, it can be seen that the five indicators of tung oil prepared by the butane-subcritical process meet the requirements of the national standard and are compared with the conventional leaching tung oil index in the literature, and the quality of the obtained products is better than that of conventional solvent-leaching tung oil. Therefore, the subcritical process has a good application prospect in the preparation of tung oil.

**Table 8.** Physical and chemical index analysis of tung oil prepared by butane-subcritical process extraction.

| Serial Number | Index | GB 8277-1987 [19] | Leaching Tung Oil [22] | Detection Value |
|---|---|---|---|---|
| 1 | Density (g/cm$^3$) | 0.9360–0.9395 | 0.9384 | 0.9412 |
| 2 | Iodine value (g/100 g) | 163–173 | 161.09 | 169.55 |
| 3 | Acid value (mgKOH/g) | 3.0–7.0 | 2.78 | 2.03 |
| 4 | Saponification value (mgKOH/g) | 190–195 | 192.15 | 194.33 |
| 5 | Refractive index | 1.5185–1.5225 | 1.5188 | 1.5321 |

### 4. Conclusions and Discussions

In this study, butane-subcritical process technology is applied, for the first time, to the preparation of tung oil, and the process of preparing tung oil from the elements of the extraction mechanism and process parameter optimization is elaborated. It was found that the preparation of tung oil by subcritical extraction technology has obvious advantages, and the experimental results provided certain theoretical support and technical support for the efficient preparation of tung oil, which leads to the following conclusions:

(1) By simulating the subcritical n-butane/tung oil mutual dissolution equilibrium model, the miscible dynamic equilibrium of tung oil in subcritical n-butane was studied under the temperature range of 35–50 °C and the extraction equilibrium time of 40 min. The solubility of tung oil in subcritical n-butane was determined, and the equilibrium solubility and process temperature were correlated with the improved Chrastil model, and an association formula suitable for the miscible equilibrium of subcritical n-butane and tung oil was constructed. $S = (-0.00125T + 0.94692)^{38.83} exp\left(\frac{-9816.15}{T} + 58.76\right)$, $a = \frac{\Delta H}{R} < 0$. It is shown that the dissolution of tung oil in subcritical n-butane is an exothermic process.

(2) The BBD module in data processing software Design-Expert 12.0 was used to design experiments at the L17 (3,3) level to obtain a quadratic regression equation model: $Y = 67.60 + 0.3275 \times A + 0.6850 \times B - 0.5775 \times C - 0.085 \times AB + 0.355 \times AC - 1.62 \times BC - 1.93 \times A^2 - 3.00 \times B^2 - 4.47 \times C^2$. P was less than 0.0001, $R^2$ was 0.9776, and $R^2$ adj was 0.9487. This showed that the fitting model was significant, and the fitting degree was high, which can accurately predict the response value of tung oil extracted from subcritical n-butane fluid under different conditions. The best prediction parameters were obtained by the regression equation: A (extraction time) was 43.77 min, B (extraction temperature) was 42.98 °C, C (material particle size) was 38.88 mesh. The model predicted the optimal extraction rate of 67.437%; in order to facilitate the operation of the experiment in the actual process, the parameters were adjusted: the extraction temperature was 43 °C, the extraction time was 44 min, and the raw material particle size was 39 mesh. Under these conditions, the experimental value was finally $67.12 \pm 0.05\%$, and the difference was not significant, which further verified the reliability of the model.

(3) Under the conditions of 35 °C, 40 °C, and 45 °C, the data of tung oil extraction rate at different times were modeled by 1stOpt data processing software: the origin software was used for mapping, and it was found that the Patricelli model accurately elucidated the kinetic process of tung oil extraction through subcritical n-butane, and $R^2$ greater than 0.99. Kinetics showed that the lipid extraction process is divided into two parts: extracting free grease and internal binding grease; free grease extraction was easy, and the internal bound grease needed to be extracted by the diffusion movement of the grease molecules to achieve the extraction of grease, so the extraction rate was low. This provides theoretical guidance for the pretreatment of raw materials in subsequent industrial applications.

(4) The extracted tung oil was analyzed by Gas Chromatography–Mass Spectrometry (GC-MS), and the α-oleostearic acid content was 74.99%, linoleic acid content was 8.83%, oleic acid content was 7.42%, palmitic acid content was 2.02%, stearic acid content was 4.35%, and the distribution of fatty acids in the references was almost indistinguishable, which verified the feasibility of preparing high-quality tung oil by the subcritical process. The physical and chemical indexes of tung oil prepared by the subcritical process were analyzed; the five indicators examined met the requirements of the GB 8277-1987 standard, and when it was compared with the tung oil leached from conventional solvents as described in the literature, it was found that the tung oil prepared by the subcritical process was of excellent quality.

Through this study, it was found that subcritical n-butane fluid extraction has good development potential in the field of tung oil preparation from the extraction mechanism

to expound the process of free oil extraction and internal combined oil extraction, for the purpose of guiding the production of oil. Compared with the traditional process, it not only greatly improved the yield of oil, but also, because the remaining cake meal was not overheated, the active substance was retained, providing high-quality raw materials for the next step of scientific research and development and providing theoretical guidance for the pretreatment of raw materials in subsequent industrial applications.

## 5. Patents

This section is not mandatory but may be added if there are patents resulting from the work reported in this manuscript.

**Author Contributions:** Conceptualization, Q.L.; methodology, A.Z., Z.Z. and X.L.; software, D.K. and Z.Z.; validation, Z.X. and W.D.; formal analysis, Z.Z.; investigation, Z.Z.; resources, A.Z.; data curation, X.L.; writing—original draft preparation, Z.Z. and X.L.; writing—review and editing, Z.Z. and X.L.; funding acquisition, A.Z. and Z.X. All authors have read and agreed to the published version of the manuscript.

**Funding:** This research was funded by the Changsha Functional Oil Technology Innovation Center (KH2101007); Hunan Forestry Bureau Outstanding Training Research Project (XLK202108-2); Hunan Forestry Science and Technology Innovation Fund Project (XLK202101-1).

**Institutional Review Board Statement:** Not applicable.

**Informed Consent Statement:** Not applicable.

**Data Availability Statement:** Data are contained within the article.

**Acknowledgments:** The authors wish to thank Hunan Academy of Forestry and Xiangtan University for providing the necessary facilities to accomplish this work.

**Conflicts of Interest:** The authors declare no conflict of interest.

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
