# Peer review of "Optimization of the Subcritical Butane Extraction of Tung Oil and Its Mechanism Analysis"

_processes, doi:10.3390/pr10061071_

Round 1

Reviewer 1 Report

This manuscript has been finished to review but this study seems to have not many literatures to provide or highlight the achievements. 

this manuscript shows many equations to calculate some values but most of equations do not show references. The authors seems to establish the experimental model but in my opinion, the advantages of this experimental model does not be showed. 

in my opinion,  the references should be added in every equations (the authors should re-check), in addition, the authors should indicate the advantages of its mechanism analysis. maybe this manuscript can become complete.

in addition, this manuscript has supporting materials but in whole review process, this supporting material disappear.

Author Response

Dear respected Ms. Cady Chang,

Thank you very much for handling our manuscript and giving us an opportunity for revision. Now we are pleased to submit our revised article entitled “Optimization of subcritical butane extraction of tung oil and its mechanism analysis” (Manuscript ID: processes-1721185).

A point-by-point response to reviewers’ comments has been attached and all the modifications were used the “Track Changes” function in the revised manuscript. We appreciate the critical comments from reviewers and we hope that all the questions have been satisfactorily addressed. We believe that our novelty and scientific merits are presented in a clearer manner in the revised manuscript for potential publication in processes.

Kindest regards,

Dr. Aihua Zhang, representative of all of the authors

State Key Laboratory of Utilization of Woody Oil Resource, Hunan Academy of Forestry, Changsha 410004, China

Email address: [email protected]

Reviewer 1:

This manuscript has been finished to review but this study seems to have not many literatures to provide or highlight the achievements. 

Response: Thanks for your critical comments and suggestions, which helped us to improve the scientific quality of this paper.

  1. This manuscript shows many equations to calculate some values but most of equations do not show references. In my opinion, the references should be added in every equations (the authors should re-check).

Response: Thanks for your valuable comments and suggestions. We have supplemented three crucial references([16]、[17]、[21]), all of which are explanations of the source of the formula, and other formulas that are not supplemented can be derived or calculated from the above formulas. (Please see Page 6, Table2.1 and Page 12 Line 302).

  1. The authors seems to establish the experimental model but in my opinion, the advantages of this experimental model does not be showed. In addition, the authors should indicate the advantages of its mechanism analysis. maybe this manuscript can become complete.

Response: Thanks for your careful comments and suggestions. This paper is to study how to obtain high-quality tung oil by using subcritical method to efficiently extract. In the process of chemical separation, it is often encountered that associative substances such as carboxylic acids, yeasts, and water are encountered, and the calculation of their thermodynamic properties cannot be ignored. Due to the presence of strong hydrogen bonding in these substances. Thus various theories of chemical association are widely used to describe the non-ideality of associative fluids. Therefore, chemical association theory is used to study the phase equilibrium law of the associative fluid, and at the same time, it can also be a good transformation process of the reaction phase equilibrium. We also added reference [21] in the revised manuscript. (Please see Page 12 Line 302).

  1. In addition, this manuscript has supporting materials but in whole review process, this supporting material disappear.

Response: Thank you for pointing out the missing information of some materials. We have double checked and modified the pages or article numbers missed.

Reviewer 2 Report

The manuscript “Optimization of subcritical butane extraction of tung oil and its mechanism analysis” describes a novel butane-subcritical extraction method to produce tung oil from vernicia fordii seeds. The process parameters were optimized through response surface methodology, examining the simultaneous influence of temperature, extraction time, and raw material particle size on the oil wield. The chemical composition of the oil produced was characterized by gas chromatography-mass spectrometry.

The entire manuscript needs a thorough revision before being considered for publication because it is written carelessly. English revision is also advisable.

The abstract is too extensive and includes excessive experimental details. On the contrary, the introduction is overly short, has very few references, and the newest one is from 2014. 

The Materials and Methods and Research Methodology sections refer to an infrared analysis of tung oil (lines 72, 73 and 165 to 168). However, there is no mention of the purpose of this analysis, and no results are shown in the Results and Analysis section. 
Concerning the statistical analysis, some equations are so widely used that they could be omitted, and it is unclear why only the extraction rate has an error associated.
In the presentation of the results: Figure 3.4 is redundant and could be combined with Figure 3.1; Do the values presented in Table 3.6 correspond to tung oil from butane subcritical extraction or leaching extraction? What does “detection value” stands for?
Finally, the main text contains various undefined acronyms, and there are several unfilled template sections by the end of the manuscript. Namely: Patents; Supplementary Materials; Author Contributions; Data Availability Statement; Acknowledgments; Conflicts of Interest; Appendix A; Appendix B.

Below is a list of questions from reading throughout the main text, illustrating why it needs rewriting:

Lines 15/16 – What does “analyzed by gas hydrangea combiner” means?

Lines 52/53 – What is “6# solvent”?

Line 84 - What is a “a CNC standard inspection”?

Lines 187, 192, 193, 217, 272 and 315 - “miscibility”, not “misclubility” or “miscility

Line 196 – What does “Changes in the quality of parvovirus oil in butane fluid” means?

Line 485 – What is “The temperament spectrum of tung oil”?

Line 487 - What is a “gas panelizer”?

Line 544 – What does “The extracted tung oil was gastrically analyzed” means?

A few suggestions regarding the abstract:

Lines 15/16 – “The fatty acid composition of tung oil was analyzed by gas hydrangea combiner…”. Please rename the technique adequately. According to the results, the analysis was performed by Gas chromatography–mass spectrometry (GC-MS)

Line 16 – “α-tung acid”. Could this be replaced by a more scientific name, like “α-oleostearic acid (α-tung acid)”?

Lines 17/18 - Please separate the two sentences. For example, rephrase from “...acid content of 4.35%, through the analysis of the oil sample obtained 5 indicators, it can be seen that...” to “…acid content of 4.35%: Through the analysis of the oil sample obtained, five indicators show that…”

Line 19 – “By simulating the subcritical n-butane/tung…” instead of “By simulated n-C4/tung …”

Line 20 – Define the acronym n-C4 “tung oil in subcritical n-butane (n-C4) was”

Line 21 – “at temperatures in the range of 35-50°C” instead of “at a temperature of 35-50°C”

Line 22 – “…, with a coefficient of determination (R2) greater than 0.99” instead of “…, and the R2 greater than 0.99”.

Lines 23/24 – The Chrastil model (equation) should be removed from the abstract.

Line 27 – Consider changing “…design. Used 1stopt data processing software…” to “design. Using 1stOpt data processing software…”. (1stopt appears two more times in the main text. It should be 1stOpt.)

Line 29 – “subcritical n-C4 with R2 greater than 0.99.” instead of “subcritical n-C4 and the R2 greater than0.99.”

Author Response

Dear respected Ms. Cady Chang,

Thank you very much for handling our manuscript and giving us an opportunity for revision. Now we are pleased to submit our revised article entitled “Optimization of subcritical butane extraction of tung oil and its mechanism analysis” (Manuscript ID: processes-1721185).

A point-by-point response to reviewers’ comments has been attached and all the modifications were used the “Track Changes” function in the revised manuscript. We appreciate the critical comments from reviewers and we hope that all the questions have been satisfactorily addressed. We believe that our novelty and scientific merits are presented in a clearer manner in the revised manuscript for potential publication in processes.

Kindest regards,

Dr. Aihua Zhang, representative of all of the authors

State Key Laboratory of Utilization of Woody Oil Resource, Hunan Academy of Forestry, Changsha 410004, China

Email address: [email protected]

Reviewer 2:

The manuscript “Optimization of subcritical butane extraction of tung oil and its mechanism analysis” describes a novel butane-subcritical extraction method to produce tung oil from vernicia fordii seeds. The process parameters were optimized through response surface methodology, examining the simultaneous influence of temperature, extraction time, and raw material particle size on the oil wield. The chemical composition of the oil produced was characterized by gas chromatography-mass spectrometry.

Response: Thanks for your critical comments and suggestions, which helped us to improve the scientific quality of this paper.

  1. The entire manuscript needs a thorough revision before being considered for publication because it is written carelessly. English revision is also advisable.

Response: Thank you for your valuable comments. We have endeavored to improve English writing, hoping to reach the requirement of this journal.

  1. The abstract is too extensive and includes excessive experimental details. On the contrary, the introduction is overly short, has very few references, and the newest one is from 2014. 

Response: We appreciate your valuable comments. We have modified the abstract and the introduction, and we also added three important references([9]、[12]、[13]) in the revised manuscript. (Please see Page 2, Line 48; Page2, Line 64 and Page 2, Line 73).

  1. The Materials and Methods and Research Methodology sections refer to an infrared analysis of tung oil (lines 72, 73 and 165 to 168). However, there is no mention of the purpose of this analysis, and no results are shown in the Results and Analysis section. 

Response: Thanks for your valuable suggestion. We have revised relevant part throughout the whole revised manuscript. (Please see Page 6, Line189-192).

  1. Concerning the statistical analysis, some equations are so widely used that they could be omitted, and it is unclear why only the extraction rate has an error associated.

Response: Thanks for pointing out this issue. We have omitted some of the formulas in the revised manuscript. (Please see Page 13, Line311-317) In the experimental phase of non-process optimization, the obtained data are all for the purpose of fitting and validating the selection of the model, so we directly selected the optimal data, so there was no error analysis.

  1. In the presentation of the results: Figure 3.4 is redundant and could be combined with Figure 3.1; Do the values presented in Table 3.6 correspond to tung oil from butane subcritical extraction or leaching extraction? What does “detection value” stands for?

Response: Thank you for your valuable comments. We have combined Figure 3.4 with Figure 3.1 in the revised manuscript. (Please see Page 7, Line200-203). The values presented in Table 3.6 correspond to tung oil from subcritical extraction. The “detection value” represents that we have tested the 5 indicators of tung oil extracted through subcritical extraction according to the national standard GB 8277-1987.

  1. Finally, the main text contains various undefined acronyms, and there are several unfilled template sections by the end of the manuscript. Namely: Patents; Supplementary Materials; Author Contributions; Data Availability Statement; Acknowledgments; Conflicts of Interest; Appendix A; Appendix B.

Response: Thanks for your valuable comments and suggestions. We have modified the undefined acronyms in the revised manuscript.

  1. Below is a list of questions from reading throughout the main text, illustrating why it needs rewriting:

7.1 Lines 15/16 – What does “analyzed by gas hydrangea combiner” means?

Response: Thanks for correcting this unprecise academic term. We used “Gas chromatography–mass spectrometry (GC-MS)” instead of “gas hydrangea combiner”. (Please see Page 1, Line16).

7.2 Lines 52/53 – What is “6# solvent”?

Response: 6# solvent is a kind of vegetable oil extraction solvent. Various hydrocarbon solvents contained aliphatic hydrocarbons, aromatic hydrocarbons and cyclic hydrocarbons with different components, main components were Pentane and Hexane. The melting point is 92.5℃. The boiling point is 67.5℃. It is flammable, low toxic, soluble in benzene, chlorine, acetone, carbon tetrachloride and other organic solvents, but insoluble in water. It is one of the basic organic chemical raw materials.

7.3 Line 84 - What is a “a CNC standard inspection”?

Response: Thank you for your valuable comments. We used CNC standard inspection to obtained the required material particle size through automatic screening, and we supplemented this equipment in revised manuscript. (Please see Page 3, Line 95).

7.4 Lines 187, 192, 193, 217, 272 and 315 - “miscibility”, not “misclubility” or “miscility

Response: Thanks for pointing out this issue. We have revised relevant part throughout the whole revised manuscript.

7.5 Line 196 – What does “Changes in the quality of parvovirus oil in butane fluid” means?

Response: Thanks for pointing out this issue. It means the change of the weight of tung oil in the extraction solvent butane at different temperatures, we have used the wrong word and have corrected them in revised manuscript. (Please see Page 8, Line222).

7.6 Line 485 – What is “The temperament spectrum of tung oil”?

Response: Thanks for pointing out this issue. We have corrected “The temperament spectrum of tung oil” to “The GC-MS spectrum of tung oil”. (Please see Page 24, Line516).

7.7 Line 487 - What is a “gas panelizer”?

Response: Thanks for pointing out this issue. We have corrected “gas panelizer” to “Gas chromatography–mass spectrometry (GC-MS)”. (Please see Page 24, Line518).

7.8 Line 544 – What does “The extracted tung oil was gastrically analyzed” means?

 Response: Thanks for pointing out this issue. We have corrected “The extracted tung oil was gastrically analyzed” to “The extracted tung oil was analyzed by Gas chromatography–Mass spectrometry (GC-MS)”. (Please see Page 27, Line577-578).

  1. A few suggestions regarding the abstract:

Lines 15/16 – “The fatty acid composition of tung oil was analyzed by gas hydrangea combiner…”. Please rename the technique adequately. According to the results, the analysis was performed by Gas chromatography–mass spectrometry (GC-MS)

Line 16 – “α-tung acid”. Could this be replaced by a more scientific name, like “α-oleostearic acid (α-tung acid)”?

Lines 17/18 - Please separate the two sentences. For example, rephrase from “...acid content of 4.35%, through the analysis of the oil sample obtained 5 indicators, it can be seen that...” to “…acid content of 4.35%: Through the analysis of the oil sample obtained, five indicators show that…”

Line 19 – “By simulating the subcritical n-butane/tung…” instead of “By simulated n-C4/tung …”

Line 20 – Define the acronym n-C4 “tung oil in subcritical n-butane (n-C4) was”

Line 21 – “at temperatures in the range of 35-50°C” instead of “at a temperature of 35-50°C”

Line 22 – “…, with a coefficient of determination (R2) greater than 0.99” instead of “…, and the R2 greater than 0.99”.

Lines 23/24 – The Chrastil model (equation) should be removed from the abstract.

Line 27 – Consider changing “…design. Used 1stopt data processing software…” to “design. Using 1stOpt data processing software…”. (1stopt appears two more times in the main text. It should be 1stOpt.)

Line 29 – “subcritical n-C4 with R2 greater than 0.99.” instead of “subcritical n-C4 and the R2 greater than0.99.”

Response: Thanks for your valuable suggestion. We have revised relevant part throughout the whole revised manuscript.

Round 2

Reviewer 1 Report

dear authors:

the response can solve my some doubts, and this manuscript can become complete. 

this can be acceptable.

Reviewer 2 Report

The authors have revised the manuscript thoroughly.

The quality of presentation and scientific soundness of the manuscript have improved significantly.

It would still be advisable to have some English native speaker read it.